# Analysis of Caesarean Section Rates Using the Robson Classification System at a University Hospital in Spain

**DOI:** 10.3390/ijerph17051575

**Published:** 2020-02-29

**Authors:** Rafael Vila-Candel, Anna Martín, Ramón Escuriet, Enrique Castro-Sánchez, Francisco Javier Soriano-Vidal

**Affiliations:** 1Department of Obstetrics and Gynaecology, Hospital Universitario de la Ribera, FISABIO. Crta. Corbera km 1, 46600 Valencia, Spain; 2Department of Nursing, Faculty of Nursing and Podiatry, Universitat de València, Jaume Roig, s/n, 46010 Valencia, Spain; soriano_fravid@gva.es; 3School of Medicine, Universidad Autónoma de Madrid, Spain. C/Arzobispo Morcillo 4, 28029 Madrid, Spain; annama7@blanquerna.url.edu; 4School of Health Sciences Blanquerna, Universitat Ramon Llull, C/Padilla 326, 08025 Barcelona, Spain; rescuriet@gencat.cat; 5NIHR Health Protection Research Unit (HPRU) in Healthcare Associated Infections (HCAI) and Antimicrobial Resistance (AMR) at Imperial College London, Du Cane Road, London W12 0NN, UK; e.castro-sanchez@imperial.ac.uk; 6Department of Obstetrics and Gynaecology, Hospital Lluis Alcanyis, FISABIO. Crta Xàtiva, s/n, 46800 Valencia, Spain

**Keywords:** caesarean section, Robson ten-group classification system, labor, delivery classification, Spain

## Abstract

Background: The WHO recommends the use of the Robson ten-group classification system (RTGCS) as an effective monitoring and analysis tool to assess the use of caesarean sections (CS). The present study aimed to conduct an analysis of births using the RTGCS in La Ribera University Hospital over nine years and to assess the levels and trends of CS births. Methods: Retrospective study between January 1, 2010, and December 31, 2018. All eligible women were allocated in RTGCS to determine the absolute and relative contribution made by each group to the overall CS rate; linear regression and weighted least squares regression analysis were used to analyze trends over time. The risk of CS of women with induced versus spontaneous onset of labor was calculated with an odds ratio (OR) with a 95% CI. Results: 16,506 women gave birth during the study period, 19% of them by CS. Overall, 20.4% of women were in group 1 (nulliparous, singleton cephalic, term, spontaneous labor), 29.4% in group 2 (nulliparous, singleton cephalic, term, induced labor or caesarean before labor), and 12.8% in group 4 (multiparous, singleton cephalic, term, induced or caesarean delivery before labor) made the most significant contributions to the overall rate of CS; Conclusions: In our study, Robson Groups 1, 2, and 4, were identified as the main contributors to the hospital’s overall CS rate. The RTGCS provides an easy way of collecting information about the CS rate, is a valuable clinical method that allows standardized comparison of data, and time point, and identifies the groups driving changes in CS rates.

## 1. Background

There is growing international concern about the increased use of caesarean sections (CS), particularly in high-income countries [1]. Caesarean procedures performed in the absence of a clinical justification do not reduce maternal or infant death rates if carried out at a rate higher than 10%–15% [2]. The unjustified, excessive use of clinical procedures can lead to an ever-increasing therapeutic cascade of avoidable interventions [3] and become life-threatening in the present or future pregnancies for both the women and children [4]. The worldwide rise in CS rates has become a growing public health concern and a cause for debate due to potential maternal and perinatal risks, cost issues, and inequity in access [5].

There is a high degree of variability in the reported crude rates of CS performed in different countries and regions, and there are often even significant differences between hospitals within a single region. The highest caesarean rates are observed in the Dominican Republic (56.4%), Brazil (55.6%), and Egypt (51.8%), with Africa (7.3%) showing the lowest proportion of these procedures [1]. In most European countries, the rates are about 25% to 35% [5]. In Spain, the average CS rate reported across the 17 autonomous communities, the governing entities independently responsible for health care [6] and for deploying health resources to serve the needs of their local populations, was found to be 24.5% in 2015 [7,8]. However, due to the decentralized structure of the health system, there is no nationally established system to monitor the use of caesarean procedures.

Achieving reductions in maternal and infant morbidity and mortality are, among others, the objectives promoted by the World Health Organization (WHO) for 2030. One of the suggested ways to meet this goal consists of avoiding clinically unnecessary caesareans [9]. However, the challenge is to keep CS rates low while ensuring safe outcomes for mothers and infants [4]. One of the main referred difficulties was the lack of a classification tool that would be feasible to be used internationally, to allow audit feedback and setting an optimal CS rate over countries. To address this gap, in 2001, Robson et al. proposed an overall classification method that facilitates an understanding of the rate of CS in a center and makes it possible to identify key subpopulation groups, all in order to inform measures aimed at preventing unnecessary procedures [10,11,12].

The WHO has proposed the use of the Robson ten-group classification system (RTGCS) as the global standard, as this classification method allows for the analysis of changing trends over time, makes it possible to compare differences between centers and shed light on how changes in clinical practice can optimize caesarean rates, thus ensuring excellence in maternal and perinatal care [13]. The more in-depth analysis offered by this method allows us to examine issues such as which groups of women and which obstetric populations are most likely to undergo caesarean sections, information that can point us toward the kinds of interventions that might help reduce the rates of caesarean sections, when and where such reductions are desirable [14].

The present study aimed to conduct an analysis of births using the RTGCS in La Ribera University Hospital over nine years and to assess the levels and trends of CS births.

## 2. Methods

### 2.1. Design, Population, and Sample

An observational study was conducted of births at La Ribera University Hospital (LRUH) (Valencia, Spain) from January 1, 2010, to December 31, 2018. The data were retrospectively extracted from the electronic birth records of women included over this period.

The health department of La Ribera has a population of 250,000 inhabitants and a yearly average of 1700 births. The hospital manages births from week 34, as below this gestational age, women are referred when appropriate to the reference hospital for neonatal unit support if required. The study’s population included women giving birth during the study period to live babies after at least 24 weeks gestation, or to a liveborn baby weighing at least 500 g.

Our study applied the ethical principles for medical research established in current Spanish legislation and was approved by the LRUH Research Commission’s Research Ethics Committee (#134-19). Considerations such as confidentiality and full information were extended to all participants.

### 2.2. Data Collection Tools

We used the RTGCS to categorize all women giving birth with 24 weeks’ gestation or longer during our study period [15]. Table 1 shows the definitions of each group.

All women were contemporaneously classified using the five obstetric characteristics described in the RTGCS (number of fetuses, parity, fetal presentation, the onset of labor, and gestational age), without requiring the indication for CS (Figure 1). All categories were totally inclusive and mutually exclusive.

A training session was conducted to introduce the implementation manual to the staff responsible for data collection. The interpretation of the data collected in the report table of the Robson classification was carried out according to three main domains: quality of information, type of population, and caesarean section rate.

The five variables collected for RTGCS included obstetric history (parity and previous caesarean section), type onset of labor (spontaneous, induced, or caesarean section before the onset of labor), fetal presentation or lie (cephalic, breech, or transverse), number of neonates, and gestational age (preterm or term). Other information collected to describe population consisted of sociodemographic data and the obstetric characteristics of the pregnant women: country of origin, age, newborn’s gender, birth weight, and feeding in the delivery room (breastfeeding, formula feeding).

### 2.3. Statistical Analysis

Statistical analyses were carried out using SPSS software version 20.0 (IBM Corp. Released 2011. IBM SPSS Statistics for Windows, Armonk, NY: IBM Corp.) Frequencies and percentages were calculated for all variables. The standard deviation (x ± SD) of the mean was calculated for quantitative variables. The overall CS rate, the relative size of each group, the CS rate within each group, and each group’s relative and absolute contribution to the overall CS rate were calculated over the study period. The relative size of each of the 10 groups was calculated by dividing the number of births in each group by the total number of births in the obstetric population and expressing it as a percentage. The CS rates were calculated by dividing the number of CS by the total number of births in each group and expressing this figure as a percentage. Finally, the percentage contribution made by each group to the overall CS rate was calculated by dividing the number of CS in each group by the total number of births in the obstetric population.

Linear regression was performed to determine the trend over time in the number of deliveries. Weighted least squares (WLS) regression was used to analyze the trends in CS over time, weighted for the total number in each ten-croup classification system group in that year. The Chi-square test was used to analyze the statistical significance of the differences in numbers of CS between the different groups. In order to calculate the risk of CS of women with induced versus spontaneous onset of labor (groups 2 and 4 vs. groups 1 and 3), an odds ratio (OR) with a 95% CI, was calculated. The significance level was set at *p* < 0.05.

## 3. Results

In terms of the characteristics of the population, the women’s mean age was 30.7 ± 5.6 years; 54.4% (8977/16506) were nulliparous, and 81.9% (13521/16506) were natives of Spain. Of the newborns, 51.4% (8097/15764) were males. The mean birth weight of the infants was 3273 ± 518 g. In terms of the feeding method employed after childbirth, 73.3% (12100/15988) of women chose to breastfeed their infants, with nulliparous women displaying a higher rate in this regard (78.1% [6792/8687]; *p* < 0.001).

The total number of births over the nine years covered by this study was 16,506, and the rate of caesarean sections performed over the period was 19% (Table 2). There was an increase in the CS rate from 18.4% in 2010 to 20.8% in 2018 by 0.8% (95% CI -0.79-0.86) annually (*p* < 0.001). There was a decrease in the total number of births over the time period (mean difference between 2010/2018: −279; 95% CI 265-292; *p* < 0.001).

Table 2 presents the distribution of the study’s population in the RTGCS and their relative and overall contribution to the CS rate. Figure 2 shows the absolute contribution of each group to the overall CS rate over time.

Table 3 demonstrates the trends in the proportions of women in the ten groups over time. The data show that nulliparous women, singleton cephalic, term, classified in group 1 (spontaneous labor) and 2 (induced labor or caesarean before labor) represented 48.9% of the total sample, whereas multiparous women with singleton pregnancies who had not undergone a previous CS (groups 3 and 4) were 41.6%. The relative contribution of group 1 to the global CS rate decreased, from 22.1% in 2010 to 20.5% in 2018 (reduction of 0.56% per year). On the other hand, it increased in group 2, going from 23.2% in 2010 to 34.9% in 2018 (an increase of 1.50% per year).

The most significant contribution to the overall total number of CS performed came from the women placed in group 2. We observed that group 1 was 2.2 times larger than group 2, meaning that the number of cases of spontaneous initiation of labor was higher than those of induced labor or elective caesareans among the nulliparous. In 2010, the ratio between groups 1 and 2 was 2.7:1 and in 2018, it was 1.7:1.

The relative contribution to the overall CS rate of groups 3 and 4 has reduced over the years. Group 3 has gone from 14.5% in 2010 to 8.9% in 2018 (reduction of 0.72% per year), and group 4 has gone from 16.6% in 2010 to 10.6% in 2018 (reduction of 0.69% per year). The comparison between groups 3 (multiparous, singleton cephalic, term, spontaneous labor) and 4 (multiparous, singleton cephalic, term, induced or caesarean delivery before labor) yielded a difference of an even greater magnitude, as the size of group 3 was 3.3 times that of group 4. The ratio between the sizes of groups 3 and 4 increased from 3.1: 1 in 2010 to 3.4: 1 in 2018.

The relative contribution of group 5, vaginal birth after a caesarean (VBAC) to the global CS rate decreased from 0.9% in 2010 to 0.7% in 2018 (reduction 0.81% per year).

The relative contribution of group 6 (all nulliparous women with a single breech pregnancy) to the global CS rate increased, from 6.0% in 2010 to 11.6% in 2018—an increase of 0.12% per year. In contrast, group 7 (all multiparous women with a single breech pregnancy including women with previous uterine scars) reduced, from 5.1% in 2010 to 2.7% in 2018—a reduction of 0.19% per year. The ratio of the size of group 6 to that of group 7 was 2.0, indicating that breech presentations were more frequent in nulliparous than in multiparous women. As overall, the ratio between groups 6 and 7 increased from 1.1: 1 in 2010 to 4.4: 1 in 2018.

The relative contribution of group 8 (all multiple pregnancies) to the global CS rate has decreased, from 5.7% in 2010 to 1.7% in 2018— a reduction of 0.24% per year. The relative contribution of group 10 to the global CS rate increased significantly, from 5.5% in 2010 to 7.5% in 2018—an increase of 0.24% per year.

We were interested in analyzing the differences between the onset of labor (spontaneous/induced) and the mode of birth (vaginal/caesarean section) in groups 1 to 4 (Figure 3). Group 1 vs. 2: 638/4891 CS from spontaneous onset of labor vs. 768/1606 CS from induced onset of labor; Group 3 vs. 4: 338/4945 CS from spontaneous onset labor vs. 290/1176 CS from induced onset of labor. We observed that the induction of labor triples the risk of having a caesarean section delivery, compared to the onset of spontaneous labor. The risk was for both, in the primiparous (group 2 vs. 1: OR = 3.6; 95% CI 3.2–4.1; *p* < 0.001) as well as in the multiparous group (group 4 vs. 3: OR = 3.6; 95% CI 3.0-4.3; *p* < 0.001).

We analyzed the indication for induction of labor in each of the groups (2a and 4a) and their relation to the mode of birth as shown in Table 4. We observed that the first cause of induction was prolonged pregnancy (PP), with a total caesarean rate of 21.1% in these induced labors, being higher in group 2. In the case of pregnancy-induced hypertension, oligoamnios and prenatal anomalies on the cardiotocographic (CTG) fetal monitoring, there was an increased rate of CS and the analysis showed differences between groups.

## 4. Discussion

The present study includes 16,506 births that were attended at the Ribera Hospital during the nine years of the study period. When studying the evolution of the activity during this period of time, we observed an annual reduction of attended births. By contrast, there is an annual increase in the global CS rate, which has meant an increase. The data show that there has been a clear trend towards increased use of CS over this period, a finding that echoes those of other studies conducted in Spain [8,14] and is aligned with worldwide trends, although the rate recorded is lower than the Spanish national average and that of other European countries [13,16]. The common overuse of CSs is a significant public health concern, and of considerable debate, due to potential maternal and perinatal risks that raise health care costs, childbirth admissions, and inequity in the access to maternity health care [17,18].

Several previous studies document a significant association between advanced maternal age (>35 years) and an increased likelihood of CS birth [19,20]. This association may be interpreted as a result of a changing social environment, but a common explanation is the pre-pregnancy morbidities associated in these cases [21]. However, as noted in Table 4, only a few women that constitute our study presented comorbidity and, none of them had the age reported, as was not required for the RTGCS. From our data, then, we would not be able to provide a hypothesis to the relation between maternal age and the risk of a CS birth.

The nulliparous population included in groups 1 and 2 (nulliparous, singleton cephalic, term), was the most significant contributor to the overall CS rate. This increase is consistent with previous studies and probably shows a link to negative consequences in women’s health in high-income countries [3,22,23,24]. The relative contribution of group 1 (spontaneous labor) to the overall CS rate decreased significantly over the study period, in line with other studies [25]. Group 2 (induced labor) makes the greatest absolute contribution to the overall CS rate (4.7% overtime period), and this is as per previously published [26]. The CS rate for this group increased significantly over the study period from 23.2% in 2010 to 34.9% in 2018. These rates of CS are similar to those that have been previously reported in studies examining CS rates in European countries using the RTGCS [22,27]. Previous reports have described that a ratio of less than 2:1 between the sizes of groups 1 and 2 may reflect a high incidence of induction and CS before labor [11]. In 2010, this ratio was 2.7:1 and in 2018, 1.7:1. Furthermore, during the analyzed period, the CS rate increased in group 2, thereby contributing significantly to the overall increased CS rate.

The relative contribution to the overall CS rate from groups 3 and 4 (multiparous, singleton cephalic, term) has been reduced over the years, as observed in other studies [27]. Even though the relative contribution of group 4a (induced multiparous women) to the global CS rate has also decreased significantly, it ranks as the third group in terms of relative contribution to the global CS rate in the study. The group sizes for groups 3 and 4 has increased, from 3.1: 1 in 2010 to 3.4: 1 in 2018, and this might explain the decrease in the number of inductions in multiparous women. In addition, there was an increase in the number of multiparous women with spontaneous onset of labor, being able to explain the decrease in the number of caesarean sections in group 4a during the study.

Once the main contributors to CS rates are identified, the next steps should be focusing on potential interventions to prevent the further CS rise. The single most common indication for induction in groups 2a and 4a was PP and, together with all types of PROM except meconium-stained liquor, they represent almost half of all indications for induction of labor. We also noticed that, in 16.3% of the medical records, the cause for an induction was not stated. In addition, in our results and as per other studies [28], we observed that induction of labor, both in nulliparous and multiparous women, increases the risk of having a caesarean section. Until the 2018 release of a large trial regarding labor induction versus expectant management [29], evidence suggested that induction of labor without medical indication was associated with an increased rate of caesarean birth [17]. In order to reduce the number of unnecessary caesarean sections, it is very relevant to assess the medical need for induction of labor appropriately and document this promptly [24].

In contrast with other studies, we observe that the contribution of group 5 (women with singleton cephalic full-term pregnancy, who have undergone at least one caesarean section) to the overall CS rate was smaller than in other countries like France [23], UK [30], and Canada [31], and has been decreasing during the study period. However, it was greater than Ireland [27], Norway, or Sweden [30]. Furthermore, our hospital has adopted new guidelines for clinical practice that recommend offering to women, who meet optimal clinical conditions with a singleton pregnancy of cephalic presentation at 37 + 0 weeks or beyond and who have had a single previous lower segment caesarean delivery, the option of having a vaginal birth [32]. This new policy might have contributed to a significant decrease in the rates of caesarean sections among the members of this group. Furthermore, significant differences between countries in VBAC rates may suggest different obstetrical care practices, some of them facilitating the increase on VBAC rates.

Lastly, groups 8 (multiple pregnancies) and 10 (premature births) have an expected contribution of the general CS rate similar to that reported by Robson and other authors [4,11,23,31,33,34].

There is a need to analyze the possible causes of the global steady growth that has been observed in the overall rate of caesarean sections. To this end, it is important to seek out classification systems that will allow us to make comparisons between different health care systems [1,35]. An earlier systematic review comparing different classification methods concluded that the Robson classification is optimal for monitoring CS [1], and the WHO has recommended adopting the Robson classification as the global standard tool for monitoring CS [4]. The application of the Robson model is a critical step in the efforts to optimize the use of these procedures, as it helps identify, analyze, and shed light on how these interventions are employed among specific, relevant groups at a given institution. It is documented that the RTGCS is a valuable clinical method that allows standardized comparations of data across countries and can be used as a common starting point to audit induction of labor and caesarean deliveries routinely [13,22,36]. Furthermore, the use of the classification system applied in this study across Spanish hospitals would help to bring awareness on each hospital performance, facilitate the comparison between hospitals and regions and align the maternity services with the current evidence and every hospital specific need. After this study was conducted, regular audits and feedback using the Robson classification system were implemented in our hospital in order to identify issues with existing practice to improve the overall quality of care.

It is essential to avoid unnecessary interventions in childbearing women, and at the same time, ensure that those interventions that are necessary take place [37]. Every effort must be made to perform these procedures on the women that truly need them rather than merely attempting to reach a given optimal rate [13]. With this perspective in mind, it is even more important to apply suitable methods to monitor and assess the results of these kinds of interventions in order to identify when and where they are overused, mainly when they are performed on healthy women who are not deemed to be at risk. The maternity team at the hospital, including the obstetric and midwifery team [17], studied here, conducts a daily, in-depth review of every caesarean section performed on the previous day in order to assess whether the clinical indications followed met the standards set out in the institution’s protocols and to provide feedback to the healthcare professional involved.

Due to the complexity of the different interconnected factors that influence the rising CS rates, interventions aimed to reduce unnecessary CS have only shown moderate success to date [1]. Any increases in obesity, age, and nulliparity among populations of women are not enough to explain increases. Addressing the non-medical reasons that drive caesarean sections, therefore, is key to reducing inappropriate use [38]. In accordance to Vogel et al. [22], factors associated with higher rates of vaginal births may include firm policies on CS due to maternal request, cultural or social pressure, differences in the legal framework for medical litigation, and strategies favoring home births, midwifery-led continuity models of care and approach to birth [39,40]. High-quality research is needed in the future to evaluate multicomponent and locally tailored interventions addressing women’s and health professionals’ demands as well as the health system when attempting to design and implement interventions aiming at reducing the number of unnecessary CS [41].

### Strengths and Limitations

The strengths of the study include the fact that few studies in Spain have analyzed the caesarean rate in a single facility. Moreover, the sample was collected rigorously, and a sample size that was large and sufficient for the estimations made. It, therefore, provides a valuable addition to the existing evidence as it provides a successful application of the Robson classification to analyze the CS rate in a setting like a tertiary hospital and the results could be compared with other hospitals or regions. Besides, the results obtained using the Robson classification method confirm the quality of the data collected under the guidelines set out by the method’s author for this purpose.

Among the limitations of this study is the possible existence of recording errors in medical records. Any errors that were detected were analyzed by the research team and recoded to reflect the data on the obstetric process collected in the medical record.

## 5. Conclusions

In our study, the main contributors to the overall CSs performed came from Robson groups 1, 2, and 4. Efforts to reduce the overall CS rate must focus on reducing the initial CS rate (groups 1 and 2). Conducting a review of the indications for inducing labor might be one of the keys to achieving a decrease in the number of caesarean sections performed at this institution. The worldwide increase in the rate of caesarean section over the past few decades has made evident the need to formulate and apply a classification system (such as the 10-group Robson method) that makes possible a comparison of the caesarean rates at different hospitals. Such a system can be used to identify the groups displaying the most significant growth in the frequency of these procedures so as to act to stem these increases and provide an easy way of collecting information about CS rate.

## Figures and Tables

**Figure 1 ijerph-17-01575-f001:**
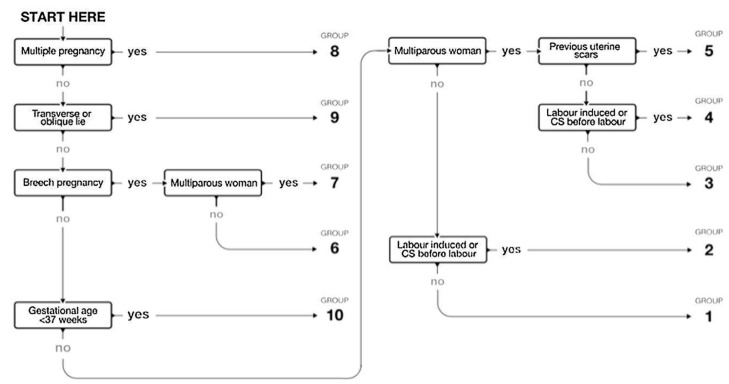
Flow chart for the Robson ten-group classification system (RTGCS). Source https://www.who.int/reproductivehealth/publications/maternal_perinatal_health/robson-classification/en/ [15].

**Figure 2 ijerph-17-01575-f002:**
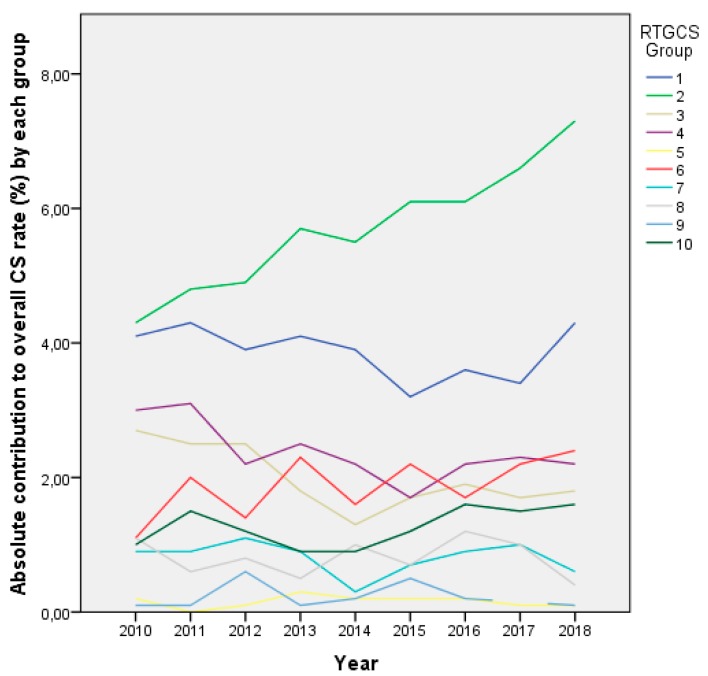
Absolute contribution to the overall caesarean section (CS) rate of each group from 2010 to 2018. RTGCS = Robson ten-group classification system.

**Figure 3 ijerph-17-01575-f003:**
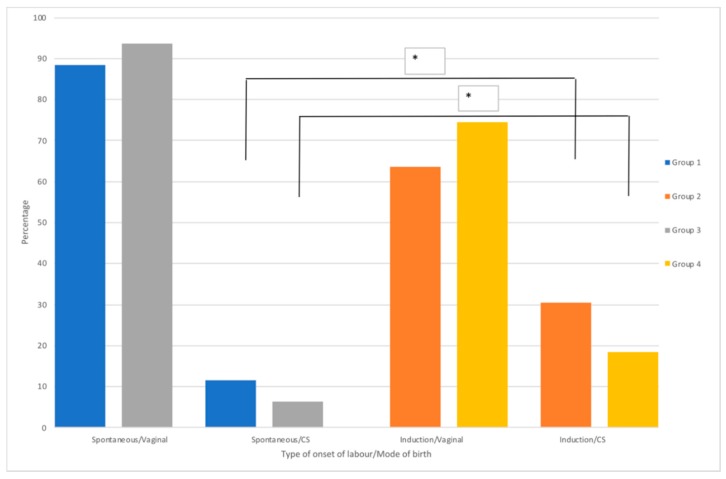
Distribution of type onset of labor by mode of birth between groups 1 vs. 3, and 2 vs. 4 in the study period. Odds ratio analyses (* *p* < 0.001).

**Table 1 ijerph-17-01575-t001:** Group description of Robson’s classification system.

1 Nulliparous, singleton cephalic, ≥37 weeks, spontaneous labor. 2a Nulliparous, singleton cephalic, ≥37 weeks, induced labor. 2b Nulliparous, singleton cephalic, ≥37 weeks, or caesarean delivery before labor. 3 Multiparous, singleton cephalic, ≥37 weeks, spontaneous labor. 4a Multiparous, singleton cephalic, ≥37 weeks, induced labor. 4b Multiparous, singleton cephalic, ≥37 weeks, caesarean delivery before labor. 5 Previous caesarean delivery, singleton cephalic, ≥37 weeks, spontaneous labor or induced labor or caesarean delivery before labor. 6 All nulliparous singleton breeches, spontaneous labor or induced labor or caesarean delivery before labor. 7 All multiparous singleton breeches (including previous caesarean delivery), spontaneous labor or induced labor or caesarean delivery before labor. 8 All multiple pregnancies, spontaneous labor or induced labor or caesarean delivery before labor. 9 All abnormal singleton lies (including previous caesarean delivery but excluding breech), spontaneous labor or induced labor or caesarean delivery before labor. 10 All singleton cephalic, ≤36 weeks (including previous caesarean delivery), spontaneous labor or induced labor or caesarean delivery before labor.

**Table 2 ijerph-17-01575-t002:** Distribution of the study’s population according to Robson’s ten-group classification system, relative and overall contribution to the CS rate (*n* = 16,506).

University Hospital of la Ribera				Period: January 2010 to December 2018		
Group	Total Number of CS in Each Group	Total Number of Women in Each Group	Group Size	Group CS Rate	Absolute Group Contribution to Overall CS Rate	Relative Contribution of the Group to Overall CS Rate
1	638	5529	33.5%	11.5%	3.9%	20.4%
2	921	2527	15.3%	36.4%	5.6%	29.4%
2a	768	2374	14.4%	32.4%	4.7%	24.5%
2b	153	153	0.9%	100.0%	0.9%	4.9%
3	338	5283	32.0%	6.4%	2.0%	10.8%
4	402	1578	9.6%	25.5%	2.4%	12.8%
4a	290	1466	8.9%	19.8%	1.8%	9.3%
4b	112	112	0.7%	100.0%	0.7%	3.6%
5	23	118	0.7%	19.5%	0.1%	0.7%
6	304	320	1.9%	95.0%	1.8%	9.7%
7	136	158	1.0%	86.1%	0.8%	4.3%
8	133	229	1.4%	82.1%	0.8%	4.2%
9	35	35	0.2%	100.0%	0.2%	1.1%
10	222	729	4.4%	27.7%	1.2%	6.4%
**Total**	**3132**	**16,506**	**100.0%**	**19.0%**	**19.0%**	**100.0%**

CS = caesarean section; Group size (%) = n of women in the group/total N women delivered in the hospital x 100; Group CS rate (%) = n of CS in the group/total N of women in the group x 100; Absolute contribution (%) = n of CS in the group/total N of women delivered in the hospital x 100; Relative contribution (%) = n of CS in the group/total N of CS in the hospital x 100.

**Table 3 ijerph-17-01575-t003:** Trends in the proportions of women in the RTGCS over time (weighted least squares regression).

Group	Change in Overall % Per Year	95% CI	*p*-Value
1	−0.56	−0.58, −0.55	0.001
2	1.50	1.48–1.53	0.001
3	−0.72	−0.74, −0.71	0.001
4	−0.69	−0.71, −0.66	0.001
5	−0.81	−1.13, −0.50	0.001
6	0.12	0.11–0.22	0.031
7	−0.19	−0.25, −0.13	0.001
8	−0.24	−0.31, −0.17	0.001
9	0.87	0.24–0.97	0.269
10	0.24	0.21–0.27	0.001

CI: confidence interval.

**Table 4 ijerph-17-01575-t004:** Distribution of indication for induction of labor in groups (2a and 4a) and their relation to the mode of birth (*n* = 3840), Chi-square analyses.

Induction of Labor Indication			Group (*n* = 3840)								
			2a (n = 2374)				4a (n = 1466)				
			CS (n = 768)		Vaginal (n = 1606)		CS (n = 290)		Vaginal (n = 1176)		
	n	% Total Row	n	% col	n	% col	n	% col	n	% col	*p*-Value
Anomalies on the CTG	193	5.0%	63	8.2%	54	3.4%	23	7.9%	53	4.5%	0.001
Prolonged pregnancy	812	21.1%	177	23.0%	341	21.2%	57	19.7%	237	20.2%	<0.001
Polyhydramnios	72	1.9%	17	2.2%	19	1.2%	10	3.4%	26	2.2%	0.088
Pregnancy-induced hypertension	189	4.9%	68	8.8%	66	4.1%	16	5.5%	39	3.3%	0.007
Antepartum hemorrhage in the 3rd trimester	16	0.4%	2	0.3%	12	0.7%	1	0.3%	1	0.1%	0.226
Not recorded	303	7.9%	26	3.4%	133	8.3%	13	4.5%	131	11.1%	0.057
Oligohydramnios	305	7.9%	69	9.0%	130	8.1%	19	6.6%	87	7.4%	0.002
Fetal pathology	4	0.1%	3	0.4%	0	0,0%	0	0.0%	1	0.1%	0.046
Maternal pathology	52	1.4%	6	0.8%	20	1.2%	4	1.4%	22	1.9%	0.482
Anhydramnios	22	0.6%	0	0,0%	15	0.9%	0	0.0%	7	0.6%	-
Favorable cervix	145	3.8%	14	1.8%	57	3.5%	8	2.8%	66	5.6%	0.135
Small for gestational age	113	2.9%	21	2.7%	59	3.7%	7	2.4%	26	2.2%	0.573
Latent phase	14	0.4%	3	0.4%	5	0.3%	1	0.3%	5	0.4%	0.393
PROM not described	355	9.2%	55	7.2%	164	10.2%	28	9.7%	108	9.2%	0.327
PROM < 12H	14	0.4%	2	0.3%	6	0.4%	0	0.0%	6	0.5%	0.186
PROM > 12H < 18H	127	3.3%	23	3.0%	46	2.9%	8	2.8%	50	4.3%	0.011
PROM > 18H < 24H	131	3.4%	20	2.6%	62	3.9%	9	3.1%	40	3.4%	0.422
PROM > 24H	390	10.2%	83	10.8%	178	11.1%	31	10.7%	98	8.3%	0.112
Meconium stained liquor	272	7.1%	48	6.3%	110	6.8%	24	8.3%	90	7.7%	0.085
Suspected macrosomia	62	1.6%	10	1.3%	22	1.4%	7	2.4%	23	2.0%	0.485
Intrauterine growth restriction	153	4.0%	34	4.4%	69	4.3%	15	5.2%	35	3.0%	0.708
Gestational diabetes	79	2.1%	21	2.7%	30	1.9%	7	2.4%	21	1.8%	0.150
Abnormal dopplers	17	0.4%	3	0.4%	8	0.5%	2	0.7%	4	0.3%	0.793

CS: caesarean section; CTG: cardiotocographic fetal monitoring; PROM: prolonged rupture of membranes.

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
