# Peer review of "Analysis of Caesarean Section Rates Using the Robson Classification System at a University Hospital in Spain"

_ijerph, 2020, doi:10.3390/ijerph17051575_

Round 1
Reviewer 1 Report
I have no further comments.
Reviewer 2 Report
This article aimed to validate RTGCS for monitoring cesarean section rates, and concluded that this RTGCS is a valuable clinical method. I had the following comments.
The conclusion did not provide useful clinical implication. Table 1 should be listed as a supplemental table. The results section: There is an increase in the CS rate from 18.4% in 2010 to 20.8% in 2018 by 0.8% annually. The above 0.8% annually should be wrong. Please recheck. In Figure 2, the title in Y axis should be wrong. From the figure, CS rate was only 6% in group 2? In Table 3, the statistical methods and comparisons to generate p value should be stated. In Figure 3, the comparison is less meaningful, and statistical comparison did not stated. In Table 4, the statistical methods and comparisons to generate p value should be stated. Many obstetrics should agree that the Robson classification system is a useful method to monitor CS rate. The rationale why the authors want to validate the above universal accepted system is not stated. Overall, the statistical method for all tables should be stated as possible. I suggest that the aim, the statistical methods and results should be rewrote or re-calculated to let the readers easily understand the aim of this study.Author Response
Please see the attachment.
Thank you.

Reviewer 3 Report
A file with the commentaries is attached.

Reviewer 4 Report
Villa-Candel R et al. demonstrated the analysis of cesarean section rates using RTGCS in Spain.
Overall the manuscript is well written, and it is a very intriguing study. However, I have a concern related to this scale. Usually, the rate of the cesarean section will be increased among advanced maternal ages. However, this scale seems not to include this factor. In developed countries, including Spain, the childbearing age has become higher than before. In contrast, there may be not a specific change in developing countries. Is the RTGCS a useful method for comparison among countries with different backgrounds? Please discuss in the text.
Round 2
Reviewer 2 Report
The quality of this article is improved. I have no comments.
Reviewer 4 Report
I have any concern regarding the manuscript.
This manuscript is a resubmission of an earlier submission. The following is a list of the peer review reports and author responses from that submission.
Round 1
Reviewer 1 Report
I was pleased to revise the manuscript entitled “Analysis of caesarean section rates using the Robson classification system at a university hospital in Spain” (Manuscript Number: ijerph-603150).
The topic of this manuscript falls within the scope of the Journal. The study was approved by the research ethics committee of the study center.
The Authors performed a retrospective study aimed to report the rate of caesarean section performed at the Hospital Universitario de la Ribera between the January 1st 2010 and December 31st 2018. Moreover, the cesarean section rate was analyzed based on the ten Robson classes.
Methods and results are well reported, and conclusion is consistent with them. Nevertheless, there are different point of concern about the manuscript:
All the text needs a language revision by a native English speaker, in order to improve its readability. The Chi square test cannot be performed on proportions or percentage, but only on the absolute number of events. Please correct the statistics or the statistics description if it was an oversight. The table are very hard to read and follow, I would suggest simply them to improve readability. The main problem of the manuscript is the novelty. Although the topic is of high interest, the publication of a center delivery report does not provide new evidence on the topic. Actually, these data can be retrieved by any center with a delivery database and are of paramount importance for the improvement of the center obstetrics care as recommended by the WHO. A such report does not provide new evidence.Reviewer 2 Report
The manuscript titled “Analysis of caesarean section rates using the Robson classification system at a university hospital in Spain” analyses the caesarean section rate according to the aforementioned classification system, with recent data spanning from 2010 to 2018 collected at the University Hospital La Ribera in Spain. The descriptive study considers an important topic and is mainly well conducted. I comment on the following points:
Abstract:
I recommend commenting on the implication of the main finding in other than methodological terms.
Introduction:
p.2. paragraph 1: last sentence (“in the index or”) is unclear, please rephrase
p.2: paragraphs 3 and 4 could be combined
p.2, paragraph 6: please tell what is the broader aim of this paper e.g. to validate the use of the classification system or/and provide information about the rising CS rate in Spain/more broadly
Materials and Methods
p.2, paragraph 7: rephrase “attended”
p.4, paragraph 3: In table 3 also the relative contribution of a group is shown, for clarify/consistency it could be mentioned here too
Results:
p.4, paragraph 4: clarify/rephrase “variables displaying significant differences”
p.4, Table 2: rephrase the title
p.4. the trend in the CS rate should be mentioned in the results, it is now only mentioned in the discussion. It could even be considered to be include in the figure 2, with a secondary axis.
p.4, Table 2: Include a column “All”, describing the whole population. You may consider showing the Ns only for the whole population, and the percentages for the different groups. This would enhance the readability of the table.
p.3, Table 2: change x -> M
p.5, paragraph 1: check the share of nulliparous women, earlier in the text another % is reported
p.5, paragraphs 1-3: especially here, please make the interpretation of the table 3 more systematic, it is not clear why something is mentioned and why something is left unmentioned (incl. group inclusion criteria), differently depending on the group in question
p.5, paragraph 4: “induce labor or elective caesareans” among the nulliparous?
p.6, paragraph 2: reference Figure 2 earlier in text
p.7: justify why some groups are included in Figure 2 why others are left out. You might also consider the option of showing the absolute instead of relative contributions in this Figure, but this would require some respective changes also elsewhere in the text.
Discussion:
p.7-8: mentioned group inclusion criteria when mentioning a group
p.7, paragraph 4: can you phrase the meaning of the first sentence more explicitly
p.7-8: organize all discussion on data quality in one paragraph, unless there is a good reason (which I don’t see now) for not doing so
p.8, paragraph 2: according to figure 2 the rel. contribution of group 4 did not increase during the observation period
p.8. the authors state that the increase in the relative contribution to the CS rate of groups 2 and 4 is a more general finding. Given this, it seems unlikely that the changes in the composition of the staff at the hospital in question would be the only reason for the increasing contribution. Could you discuss other potential factors a bit more, why would e.g. induced labors be more likely to lead to SC today than before. Perhaps descriptive characteristics included in table 2 are helpful.
p.8, paragraph 3: “has increased in recent years” in this group
p.8, paragraph 4: could you specify “staff” who conducts a daily review
p.8, paragraph 4: please clarify/rephrase last sentence
p.9, paragraph 2: please clarify the second last sentence
Reviewer 3 Report
This was a large retrospective observational study to classify and analyze the cesarean rates using the Robson classification system in a tertiary referral hospital in Spain over 9 years. The aim was to determine which groups of women contribute the most to the overall institutional cesarean rates. These results can be applied to designing interventions to reduce cesarean rates. The study is methodologically straightforward. I have the following comments:
Abstract: Groups 2 and 4 had the highest group specific CS rate, but it was Groups 1 & 2 that made the largest contribution to the overall CS rate. Please revise or clarify this. Also, please include a brief description of the Groups (2 and 4 or 1 and 2 depending on revision) and include the rates of relative contribution in the abstract so that the reader does not have to read the entire manuscript to understand the results presented.
Methods, Table 1: Please include singleton in the description for Groups 6, 7, & 9.
Results: An entire paragraph and a rather large table (Table 2) is devoted to the characteristics of the population but these are not discussed further. I don’t believe these add to the objective of the paper. If the authors intended to discuss these factors in the context of resource allocation, it would be more relevant. However, given the current content, I suggest collapsing table 2 to reflect these factors by Cesarean and vaginal delivery rather than the 10 Robson groups.
Figure 2 requires axis titles (i.e. cesarean rates (%) for the y axis). The title needs to be more specific (overall rate should be overall cesarean rate) and instead of study period I would suggest over time or from 2010 to 2018. An explanation of the groups should be provided in the description of Figure 2 to aid the reader in the interpretation of the graph.
Discussion: Again, verify and/or clarify the statement that Groups 2 and 4 made the largest contribution toward CS rates. Also, in the same paragraph the authors state ‘the percentage of these women who underwent caesarean procedures increased significantly’ in reference to Groups 2 and 4 when this is only true for Group 2 according to Figure 2. This statement should be amended.